# Salivary Metabolomics for Oral Cancer Detection: A Narrative Review

**DOI:** 10.3390/metabo12050436

**Published:** 2022-05-12

**Authors:** Karthika Panneerselvam, Shigeo Ishikawa, Rajkumar Krishnan, Masahiro Sugimoto

**Affiliations:** 1Department of Oral Pathology and Microbiology, Karpaga Vinayaga Institute of Dental Sciences, GST Road, Chinna Kolambakkam, Palayanoor PO, Madurantagam Taluk, Kancheepuram 603308, Tamil Nadu, India; karthikaomfp@gmail.com; 2Department of Dentistry, Oral and Maxillofacial Plastic and Reconstructive Surgery, Faculty of Medicine, Yamagata University, Yamagata 990-9585, Japan; shigeo_ishikawa2011@yahoo.co.jp; 3Department of Oral Pathology, SRM Dental College, Bharathi Salai, Ramapuram, Chennai 600089, Tamil Nadu, India; drrajkumar22163@gmail.com; 4Institute of Medical Research, Tokyo Medical University, Tokyo 160-0022, Japan; 5Institute for Advanced Biosciences, Keio University, Yamagata 997-0811, Japan

**Keywords:** oral cancer, metabolomics, saliva, diagnosis, prognosis, machine learning

## Abstract

The development of low- or non-invasive screening tests for cancer is crucial for early detection. Saliva is an ideal biofluid containing informative components for monitoring oral and systemic diseases. Metabolomics has frequently been used to identify and quantify numerous metabolites in saliva samples, serving as novel biomarkers associated with various conditions, including cancers. This review summarizes the recent applications of salivary metabolomics in biomarker discovery in oral cancers. We discussed the prevalence, epidemiologic characteristics, and risk factors of oral cancers, as well as the currently available screening programs, in India and Japan. These data imply that the development of biomarkers by itself is inadequate in cancer detection. The use of current diagnostic methods and new technologies is necessary for efficient salivary metabolomics analysis. We also discuss the gap between biomarker discovery and nationwide screening for the early detection of oral cancer and its prevention.

## 1. Introduction

Oral cancer (OC) is a blanket term used to describe any cancer occurring in the oral cavity. In 2018, more than 350,000 new cases of OC and 170,000 deaths were recorded worldwide [1]. Tobacco usage, alcohol consumption, and human papillomavirus infection are the major risk factors for OC [2,3,4]. A recent study compared the incidence of OC in the 10 most populous countries over the past 30 years and reported declining trends in the annual age-standardized incidence rate of OC in Bangladesh, Brazil, Mexico, and the United States; however, increasing trends were observed in China, Indonesia, Pakistan, India, and Japan [5]. The 5-year overall survival rate of OC is approximately 50% [6]. To improve the prognosis and quality of life of patients, early detection of OC is essential [7].

The underlying epigenetic mechanisms and major risk factors of OC vary across countries. India accounts for one-third of the global OC cases, with 77,000 new OC cases and 52,000 related deaths annually [8]. Tobacco consumption is the main etiological factor. Most OC cases are diagnosed at advanced stages owing to delays in reporting to healthcare professionals [9]. Approximately 60–80% of the patients with OC diagnosed at late-stage Early detection can improve treatment efficacy and prognosis. Although various methods are available for screening, visual examination is the most commonly used owing to its low cost [10]. However, diagnosing lesions in the initial stage and differentiating them from inflammatory conditions remain challenging.

Despite the declining trend of tobacco use in Japan, the incidence of OC has increased [11]. Similar to the global trend, many patients are diagnosed with late-stage OC. In Japan, nationwide screening of five cancers (gastric, colon, and lung cancers for both sexes and breast and cervical cancers for women) is conducted annually or every other year. Insufficient screening is among the reasons for the increasing trend of OC. Therefore, the development of a new cost-effective screening system for OC is necessary.

Saliva is a mixture of biofluids and plays vital roles in oral homeostasis. Other functions of saliva include lubrication, digestion, buffering, taste, tooth protection, and immune defense by protecting against bacteria, viruses, and fungi. Saliva consists of various cellular and molecular components, such as transudate of the oral mucosa, desquamated oral epithelial cells, blood cells, oral bacteria, proteins, metabolites, and inorganic ions (Figure 1). Furthermore, it is mainly secreted from three major salivary glands (parotid, submandibular, and sublingual glands) and other minor glands. It also contains various components which originate from other sources, such as gingival crevicular fluid. Overall, these components make saliva an ideal biofluid for detecting various diseases.

There are several advantages of using saliva for cancer detection. First, a positive correlation has been reported between salivary and plasma metabolite levels, such as those of glucose, pyruvate, and lactate [12,13], indicating that salivary metabolites provide biological information. Second, saliva is the most readily available biofluid, and its collection requires minimal training [14]. Third, analysis of saliva samples is convenient owing to the noninfectious collection process, easy transportation, and disposable nature [15]. Fourth, the saliva metabolite profile of each individual is affected by diet compared to that of urine collected from identical individuals [16]. Therefore, several cancer biomarkers have been identified using salivary omics technologies.

Owing to the ongoing global COVID-19 pandemic, interest in cancer detection has decreased [17]. However, saliva-based tests, such as PCR [18] and antigen-based tests [19], have become popular. Therefore, it is an opportune time for the development and distribution of saliva-based cancer tests. This article reviews the recent applications of salivary metabolomics in the identification of diagnostic and prognostic biomarkers of OCs. Since saliva is found in the oral cavity, most applications are in cancers of the oral cavity, such as oral and salivary gland cancers. Nevetheless, the biomarkers of cancers originating in organs that are far from the oral cavity are also reviewed. Finally, the current screening schemes and future perspectives of salivary metabolomics are discussed.

## 2. Applications for Cancer Biomarker Discovery

### 2.1. Diagnostic Markers for OC

Salivary metabolomics has been used to identify new metabolite biomarkers to differentiate between OC and healthy control (HC) samples using various detection devices coupled with metabolite separation technologies. Biomarkers for oral squamous cell carcinoma (OSCC), which accounts for approximately 90% of all oral malignant neoplasms, have been frequently reported [6,8]. The development and validation of multiple marker-based indexes have been conducted by several studies to distinguish OC and HC groups.

Several profiling methods to identify new biomarkers and various data analyses to utilize multiple biomarkers to discriminate between the diseased group and the HCs are currently available. The analytical instruments, discrimination methods, comparison designs, and accuracies are summarized in Table 1. The details of each study, including the accuracy of individual biomarkers, are summarized in Appendix A.

Reverse phase liquid chromatography with mass spectrometry (RPLC-MS) and hydrophilic interaction chromatography-MS (HILIC-MS) were used to analyze hydrophilic metabolites in saliva samples [20]. Orthogonal partial least squares discriminant analysis of five metabolites (propionylcholine, *N*-acetyl-L-phenylalanine, sphinganine, phytosphingosine, and S-carboxymethyl-L-cysteine) enabled the distinction between early-stage OSCC and HC samples. Capillary electrophoresis-time-of-flight MS (CE-TOFMS) was also used to quantify hydrophilic metabolites in saliva samples collected from patients with OSCC and HCs in a Japanese population [21]. Univariate analyses were used to quantify 25 hydrophilic metabolites, such as choline and urea, and revealed a substantial difference between OSCC and HC samples. Gas chromatography-MS (GC-MS) was used to analyze OSCC and HC saliva samples obtained from a South American population [22]. The results of receiver operating characteristic analysis identified 24 metabolites, such as malic acid, maltose, methionine, and inosine, for distinguishing between patients with OSCC and HCs. These studies have aided the identification of potential biomarkers of OSCC; however, the sample sizes of these studies were small (*n* < 100). Conductive polymer spray ionization MS (CPIS-MS) was used to identify biomarkers to discriminate between OSCC and premalignant lesions from HC samples using a relatively large-scale dataset (*n* = 373) [23]. These markers successfully discriminated between OC and HC samples; however, their specificity against other inflammatory diseases has not been evaluated. The markers and biomarkers that are used to detect OSCC and premalignant lesions are also useful for diagnosing subjects who require further examination by clinicians. Additionally, the use of different analytical methods with non-standardized sample collection protocol may cause heterogeneous results despite the similar study design used in these studies. Thus, the reproducibility of these identified biomarkers should be confirmed before clinical application.

### 2.2. Biomarkers That Discriminate between OC and Other Diseases

Visual inspection alone is insufficient to differentiate OC from several inflammatory diseases of the oral cavity. To address this, salivary metabolomics have been applied. Salivary metabolites collected from patients with OSCC and oral leukoplakia (OLK) and HCs were analyzed using ultra-performance liquid chromatography-quadrupole time-of-flight MS (UPLC-QTOFMS) [24]. Multiple logistic regression (MLR) models using lactic acid and valine was used to distinguish between OSCC and HC samples, whereas another model based on these two metabolites and phenylalanine was used to distinguish between OSCC and OLK samples [24]. The salivary metabolite biomarkers with considerable differences between OSCC and oral lichen planus (OLP) were reported for samples collected from a Japanese population [25]. Their results suggest that markers can be used to detect malignant transformation of OLP. A similar comparison between patients with OSCC or OLK and HCs was performed using samples collected from an Indian population [26]. However, different biomarkers were identified in these studies. One possible reason is the different analytical methods used (CE-MS in the study conducted in Japan and LC-MS in India). Therefore, identical protocol and analytical methods should be used to evaluate country-dependent variations. These biomarkers represent OC-specific metabolism and are helpful in discriminating OC from other diseases, thereby contributing to early OC diagnosis.

### 2.3. Other Biomarkers Identified Using Salivary Metabolomics

In addition to the discovery of diagnostic biomarkers, other various applications of salivary metabolomics, such as the prediction of potential treatment plans, have been explored. ^18^F-fluorodeoxyglucose positron emission tomography/computed tomography (^18^F–FDG PET/CT) is commonly used to diagnose different aspects of cancer, such as metastasis, recurrence, staging, and screening. Salivary metabolites having correlations with the PET maximum standardized uptake value of ^18^F–FDG PET/CT were reported [27]. Medication-related osteonecrosis of the jaw (MRONJ) is a severe adverse effect of bone-modifying agents used to prevent bone complications in OC with bone metastasis, and salivary biomarkers for MRONJ prediction have been identified [28]. Biomarkers to estimate radiotherapy response in head and neck cancer (HNC) have also been analyzed. The time course of the metabolomic profile was studied before and during the therapy [29].

### 2.4. Biomarkers for Other Cancers

Salivary metabolomic studies have identified diagnostic biomarkers not only for OC but also for other cancers [30]. Salivary tests for breast cancer diagnosis have been developed, and various types of molecules have been proposed as biomarkers. Volatile metabolite compounds found in breath and saliva have shown potential in breast cancer detection [31]. Non-volatile metabolites, such as amino acids, can also be used to differentiate between breast cancer and HC samples [32,33]. Non-targeted non-volatile metabolite profiling of saliva revealed the presence of 18 metabolites, such as LysoPC, related to breast cancer [34]. Seven oligopeptides and six glycerophospholipids showed breast cancer-specific differences [35]. Metabolomic profiling using total protein and antioxidant enzymes helped the detection of breast cancer in individuals without breast pathologies [36]. These studies conducted non-targeted metabolomics, and biomarkers were identified based on the comparison between saliva samples obtained from patients with breast cancer and HCs.

Targeted metabolomic analysis revealed elevated polyamine levels in saliva samples [37,38,39]. Furthermore, activated synthesis and acetylation of polyamines have been observed in various cancer cells [40,41], along with higher polyamine concentrations in the blood and urine samples of patients with cancer [42]. Thus, the elevation in the level of salivary polyamines is expected to be consistent with these changes in blood samples.

Pancreatic cancer (PC) has a high mortality rate, and the absence of early specific symptoms delays its diagnosis. Although tumor markers, such as the carbohydrate antigen 19-9 and carcinoembryonic antigen, are available, their sensitivity is limited [43]. Therefore, the development of a new diagnostic method is needed. We have previously reported the PC-specific elevation of hydrophilic metabolites in saliva, and their profiles were different from those of oral and breast cancers [30]. Asai et al. analyzed the salivary polyamine profile of patients with PC, chronic pancreatitis, and HCs and found a considerable elevation in the polyamine levels of patients with PC [44]. A positive correlation was reported between polyamine concentrations in PC tissues and urine [45]. Recently, polyamine-related aberrance was reported in PC. A transcriptomics study found that the gene expression of proteins associated with polyamine synthesis pathways is associated with the prognosis of PC [46]. In addition, polyamine concentration patterns accurately distinguished among PC, chronic pancreatitis, and HC samples [47]. These data reinforce the biological rationality of salivary biomarkers, although efforts should be directed to obtain more direct evidence.

### 2.5. Salivary Metabolomics for Oral Cavity Diseases

The specificity of OC biomarkers should be evaluated in other various inflammatory diseases in the oral cavity, such as periodontal diseases (PD). A recent meta-analysis of salivary metabolomics data on PD found increased concentrations of valine, phenylalanine, isoleucine, tyrosine, and butyrate and decreased concentrations of lactate, pyruvate, and N-acetyl groups [48]. Increased levels of amino acids and short peptides were observed in the saliva of patients with PD, suggesting activated protein degradation [49]. The upregulation of oxidative stress-related pathways, such as purine degradation, was also consistently noted in PD [50]. The overlap in the metabolitebiomarkers between PD and OC could cause misdiagnosis. The validation study included not only HC but also patients with PD to evaluate marker specificity.

## 3. Technical Challenges in Salivary Metabolomic Studies

### 3.1. Metabolite Measurement Technologies

The word metabolomics was coined by merging two terms—omics and metabolites. Therefore, it is expected to refer to an analytical method that measures all metabolites. However, no single method can be used to analyze all metabolites because of the large diversity of chemical structures of metabolites in biological samples [51]. Therefore, various methods have been developed, and each technique has its own advantages and disadvantages. Various metabolite separation and detection systems have also been used to analyze metabolites in saliva samples. 

Nuclear magnetic resonance (NMR) is the most frequently used method [52]. Compared to mass spectrometry (MS), NMR has higher reproducibility and minimal preparation for any sample type [53]. Pretreatment of the saliva, a viscous liquid, is also a simple process [54]. This feature is a definitive advantage as it minimizes the chances of causing unexpected errors. NMR has enabled identification of pattern changes in salivary metabolomic profiles, i.e., metabolic signature, to distinguish between patients with cancer and HCs. Some applications of salivary metabolomics explored using NMR include the detection of OSCC [24,55], head and neck squamous cell carcinoma [56], and glioblastoma [57]. In addition to cancer, hepatitis B infection [58], Parkinson’s disease [59], and Alzheimer’s disease [60] have been analyzed. The salivary biomarkers, saliva collection methods, and NMR methods are summarized in Appendix A.

MS is another a major metabolite detection system with high sensitivity. It consumes a small volume of samples and enables the identification and quantification of hundreds of metabolites simultaneously [53]. However, direct injection to MS cannot separate the metabolites with the same *m/z* (mass divided by charge number) value, such as leucine and isoleucine; therefore, a separation system is usually used before MS. GC-MS allows the quantification volatile compounds and the profiling of non-volatile metabolites by derivatization, which was used to analyze OSCC samples [22]. Liquid chromatography (LC)-MS has been used for both non-targeted and targeted analyses of salivary metabolites. For non-targeted analyses, hydrophilic metabolites, such as γ-aminobutyric acid, phenylalanine, valine, and lactic acid, of saliva samples of OC patients were analyzed [24]. A wide variety of metabolites, such as oligopeptides, phosphatidylcholine, and glycerophospholipids, were analyzed in the saliva samples of patients with breast cancer [34,35]. For targeted analyses, salivary OSCC biomarkers, such as choline, betaine, pipecolinic acid, and carnitine, were quantified [61]. Salivary polyamines were also analyzed as known biomarkers for breast cancer [38]. Capillary electrophoresis (CE)-MS was used for hydrophilic metabolite profiling of saliva samples of OC [21,25,27,62,63], breast cancer [39], and pancreatic cancer (PC) [44]. 

Comparisons of NMR and MS for analyzing saliva samples for OC biomarker discoveries have been previously conducted [64,65]. Both reviews claimed the necessity of standardization of sample collection and the processing of measuring data. The simultaneous use of NMR and LC-MS to analyze salivary metabolites succeeded in the coverage expansion of the observed metabolite [66], enhancing the opportunity to find biomarkers related to the focused phenotype.

### 3.2. Discrimination Methods

To identify biomarkers, conventional univariate analyses, such as the Student’s *t*-test and the Mann–Whitney test for two-group comparisons, have been used in previous studies. Additionally, multivariate analyses were frequently used to analyze the similarity and the difference of overall metabolite profiles. As unsupervised methods, principal component analysis (PCA) and hierarchical clustering analysis have been performed. For example, PCA was used to assess the relative strength of the effects of multiple factors, such as inter and intraday variations, on salivary metabolomics [67]. Clustering was used to find new subgroups of a disease group based on the observed metabolomic profiles [68]. These methods help find outliers, assess the quality of samples, and form the groups used in subsequent analyses.

To discriminate a disease group from other groups, such as OC from HC, a combination of multiple metabolite concentration patterns was used (Table 1). MLR is one of the conventional multivariable methods. It uses minimal independent metabolite sets by eliminating multicollinearity problems [25,30,62]. Lasso regression model solved the multicollinearity problem [23]. Partial least squares-discriminant analysis (PLS-DA) is also frequently used to discriminate against multiple groups, enabling the ranking of the metabolite’s contribution to the discrimination. For example, discrimination among OSCC, OLK, and HC was conducted using this method [24]. Random forest, a classification machine learning (ML) model that leverages multiple decision trees [69], and alternative decision trees [39] have also been used to discriminate a group from the others. The most important concern is rigorous validation of the generalization ability to eliminate overfitting. Cross-validation is commonly used in a single cohort, while accuracy evaluation using an independent cohort provides a more rigorous validation [23].

### 3.3. Standard Operating Protocols (SoP)

The discovery and validation of biomarkers are the initial steps to establish new screening methods. A standard protocol for working with saliva samples should be determined to enhance reproducibility. The methods for preconditioning, sample collection, storage, preprocessing, measurement, and data analysis should be standardized [70]. The effect of inter-day and intra-day variations on salivary metabolomics has been analyzed [67], and no significant salivary flow was observed in the comparisons. The stimulated saliva showed larger variations in metabolomic profiles than the unstimulated saliva. The time period between the last diet and sample collection also affected salivary metabolomic profiles [63]. As expected, longer fasting conditions before sample collection improved the discrimination ability of the OC biomarkers. Normalization of overall concentration with the total contents of amino acids decreased the variations due to fasting conditions. Stimulation of the oral cavity, for example, using tobacco and mouthwash, also affected the final results [71], with stimulated and unstimulated saliva having different metabolomic profiles [72]. Taken together, the longer the fasting period, the more consistent the use of stimulated or unstimulated saliva samples. Thus, a restriction affecting the oral cavity should be defined as part of the SoP.

The effect of storage conditions on the quantified concentration of metabolite biomarkers was also analyzed [61]. Variations between short-term storage at room temperature (up to 24 h) and long-term storage at −35 °C (up to 1 month) of four OC biomarkers, such as choline and betaine, were detected. Storage and preprocessing also affected the polyamine profiles in saliva [73]. The artificially generated noise according to the maximum variations observed during storage and preprocessing enabled the estimation of possible deterioration of discrimination abilities of the biomarkers. Such analyses would assit in the stablishment of SoP in the clinical settings.

Because it is not necessarily limited to salivary metabolomics, MS-based metabolomics required better quality control than NMR-based ones [53]. Therefore, quality assessment for each processing step and a control method were developed to normalize the quantified data to ultimately eliminate unexpected bias in multiple batch measurements [74]. Along with these standardizations, the development of an automatic pipeline is also a reasonable approach [23]. The establishment of a rigorous protocol will likely yield reproducible results; however, it may hinder the widespread use of salivary tests (Figure 2). 

## 4. Discussion

### 4.1. Current OC Screening Strategies

The diagnosis of OC at an early stage can influence the treatment plan and the patient’s chances of survival. Therefore, screening programs are conducted to lower the mortality rate. In India, limited community-based screening programs and sporadic opportunistic camp-based screenings are performed. Therefore, high levels of awareness about screening and self-examination are recommended by the National Cancer Control Program of India [75]. In particular, screening (oral visual inspection) was recommended for individuals aged >35 years belonging to the high-risk group [76]. Current screening programs are conducted every three years [77]. However, the median duration for progression from early-stage HNC to advanced stage is 11.3 months, and that from advanced stage to untreatable conditions is 3.8 months [78]. According to the recommendations of the American Cancer Society, asymptomatic patients aged 20–40 years are screened every 3 years, patients aged >40 years are screened annually, and high-risk subjects should be screened every year irrespective of age [79]. Therefore, OC screening using salivary tests could be performed at shorter time intervals than the current guidelines.

The screening programs for OC rely on visual examination. However, the procedure has several limitations, including low specificity and difficulty in identifying tumor lesions [80]. Molecular studies, identification of a distinct tumor progression model of OSCC, and robust data collection methods will enhance the effectiveness of screening programs. Additionally, unnecessary tests and costs will be reduced [77]. Saliva reflects changes in the tissue even before any symptoms manifest and has been identified as the ideal diagnostic fluid to study molecular changes in OSCC [23]. Salivary metabolomics is conducted to facilitate the use of saliva in OC biomarker identification [14]. Validating the identified metabolite markers would highlight the advantages of using saliva as a non-invasive tool for screening patients. 

### 4.2. Futuer Prospects for Salivary Metabolomics

Several problems have to be addressed before the clinical application of salivary metabolomics. First, a large-scale validation study should be conducted to evaluate the accuracy of the biomarkers. Published studies on salivary metabolomics included only case-control studies. The relationship between the study design and cohort is depicted in Figure 3. Multigene expression test for OSCC was established based on diagnostic performance in a multi-institute evaluation in three countries [81]. Such large-scale validation is required for salivary metabolomics; however, two obstacles exist. First, the high reproducibility of metabolomic profiles in human biofluid in multi-centers is still challenging [82], and the standardization of the various processes for blood and urine samples was recommended [83]. In addition to measurement, the unified protocol should be established to deal with saliva samples, such as for sample transfer and storage protocols. Second, the prevalence rate of cancer subjects was low in the actual cohort, indicating that validation requires large sample sizes to include a sufficient number of patients with OC. High-throughput and cost-effective assays to realize targeted quantification of the identified biomarkers should also be developed and validated. The multilateral merits of salivary-based screening, including clinical and economic aspects, should be evaluated.

Second, the disease specificity of biomarkers should be validated. Most published studies included only one type of cancer and thus cannot be used to evaluate the specificity of the identified biomarkers. Salivary metabolites that distinguished OSCC from OL+ have been identified [25]. These biomarkers yielded different results between OSCC and HC samples [62]. Therefore, their combinations should be used to differentiate among these groups. However, the diagnosis of precancerous lesions is also vital for the early detection of these diseases.

Third, the biological relationship between salivary biomarkers and cancers should be elucidated. Various processes, such as carbohydrate metabolism, oxidative stress-related metabolism, and nucleotide synthesis, and molecules, such as polyamines, amino acids, and lipids, have been reported as metabolite biomarkers of for [84]. We compared the metabolomic profiles of OC tumor lesions and saliva samples and identified 17 metabolites, such as amino acids, lactate, and polyamines, which showed consistent evaluation results [62]. Elevated salivary polyamine levels in patients with OC were consistently reported by several studies [61]. Ornithine is used as a substrate for the synthesis of polyamines by ornithine decarboxylase (EC 4.1.1.17). Acetylation of polyamines by the enzyme spermidine/spermine acetyltransferase (EC 2.3.1.57) resulted in the formation of *N*^1^-acetylspermine or *N*^1^-acetylspermidine [85]. These enzymes are activated downstream of several oncogenes, such as *c-MYC* and *Tp53* [86]. These oncogenes have been associated with OC promotion and progression [87,88]. Therefore, elevated levels of metabolites involved in these pathways are expected, and their use as biomarker should be explored.

The association between the characteristics of saliva and cancers originating in organs distant from the oral cavity has also been studied. Saliva collected from patients with breast cancer with *BRCA1* mutation is characterized by an enhanced antioxidant capacity and oxidative damage to proteins and lipids [89]. Isolated exosome-like microvesicles from breast cancer cells showed an altered mRNA expression profile of the salivary gland [90], which is expected to change the concentration of components secreted by the salivary gland in these patients. Salivary polyamines were altered in patients with invasive carcinoma but remained unchanged in those with ductal carcinoma in situ (DCIS) [39]. This is a reasonable finding because only invasive cancer cells secrete molecules into the tumor microenvironment [41]. However, this feature is disadvantageous as salivary polyamines cannot be used to detect DCIS. Polyamine levels were consistently elevated in both pancreatic tissues and urine samples Conventionally, only *N*^1^, *N*^12^-diacetylspermine was evaluated as a biomarker; however, a combination of various acetylated forms of urinary polyamines showed potential in screening various cancers [91]. Similar data have been reported in colon cancer cohorts [92,93]. Therefore, cancer-type specificity should be rigorously evaluated.

### 4.3. Future Perspectives on Cancer Screening Using Salivary Metabolomics

The identification of cancer biomarkers alone is not adequate. The development of a high-throughput, cost-effective assay, and combination of biomarkers with other diagnostic factors are necessary to realize effective screening and treatment. A nomogram utilizes MLR to combine multiple features linearly, thus predicting the disease status and treatment outcomes. For example, a combination of clinical characteristics and serum inflammation markers has been developed to predict overall survival in OSCC patients [94].

In Japan, an OC screening showed that an opportunistic screening system is more effective in diagnosing precancer and cancer patients than a countermeasure screening system [95]. A recent systematic review and meta-analysis concluded that the diagnostic accuracy of commonly used OC screening tests (such as conventional oral examination, vital rinsing, light-based detection, mouth self-examination, remote screening, and biomarker-based screening) was inadequate for detecting OC efficiently [96]. In addition, since screening by trained dentists and oncology specialists is expensive, the development of new technologies for the objective assessment of the risk for OC is essential. Recently, several machine learning-based data processing methods have been developed. OC detection, automated staging, and distinction between cancerous and precancerous cells by image processing are typical examples of machine learning applications [97,98]. In India, various research programs for OC that use artificial intelligence are currently underway (Figure 4).

## 5. Conclusions

Studies aiming to identify metabolite biomarkers for OC detection have been extensively conducted. In addition to conducting rigorous clinical validation studies, establishing standard operating procedures for the use of saliva samples is mandatory. An effective screening system should be developed by combining conventional and modern technologies.

## Figures and Tables

**Figure 1 metabolites-12-00436-f001:**
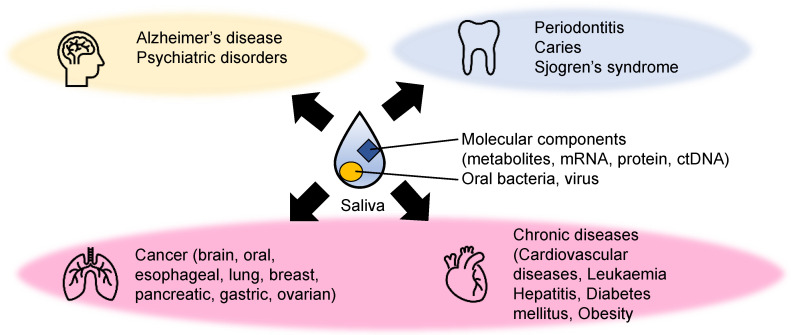
Examples of diseases detected using metabolite biomarkers. The biomarkers of mental illnesses (yellow), dental diseases (light blue), and various systematic diseases (light pink) related to metabolic abnormalities have been explored.

**Figure 2 metabolites-12-00436-f002:**
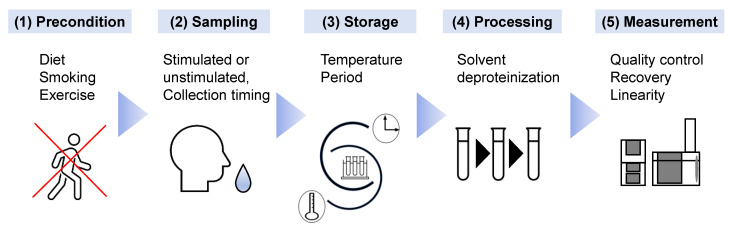
The factors influencing salivary metabolomics that require the establishment of standard operating procedures are shown. (1) As preconditions before saliva collection, the diet, smoking behavior, and intensive exercise of subjects should be restricted, as they potentially affect salivary metabolites. (2) To eliminate diurnal variation, stipulated timing of saliva collection, e.g., 9:00–11:00 am, should be implemented. The selection of stimulated or unstimulated saliva is vital. (3) The temperature and duration of storage of the saliva samples should also be standardized. (4) Chemicals used as solvents and the methodology for deproteinization should be selected accurately. (5) Quality assessment of the measurement device using quality control samples is vital to eliminate unexpected bias.

**Figure 3 metabolites-12-00436-f003:**
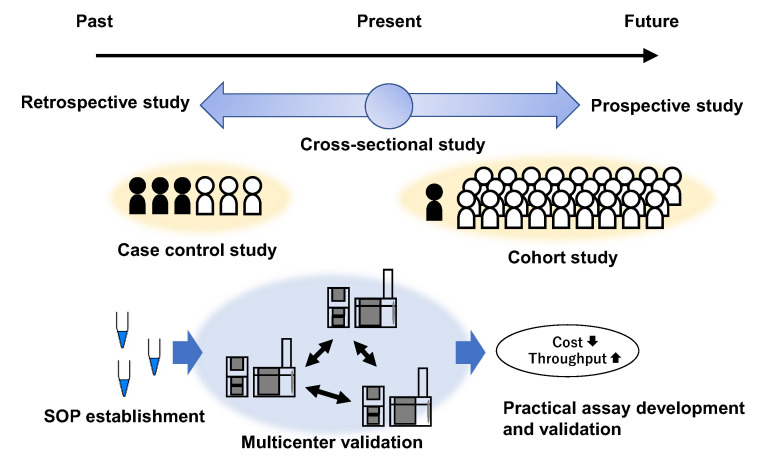
Various study designs used from salivary metabolomics to saliva-based OC test. The top layer describes the general study designs used to identify and validate diagnostic markers. The middle layer describes case-control and cohort studies. The top layer controls the number of OC and non-OC participants, whereas the middle layer includes a low rate of OC owing to the low prevalence rate in the actual cohort. The bottom layer shows the requirements of the saliva-based OC test. SoP establishment and multi-center-level validations are necessary to discover and validate biomarkers using metabolomics. The development and validation of a cost-effective and high-throughput assay that enables targeted analyses of the biomarker is required for clinical use.

**Figure 4 metabolites-12-00436-f004:**
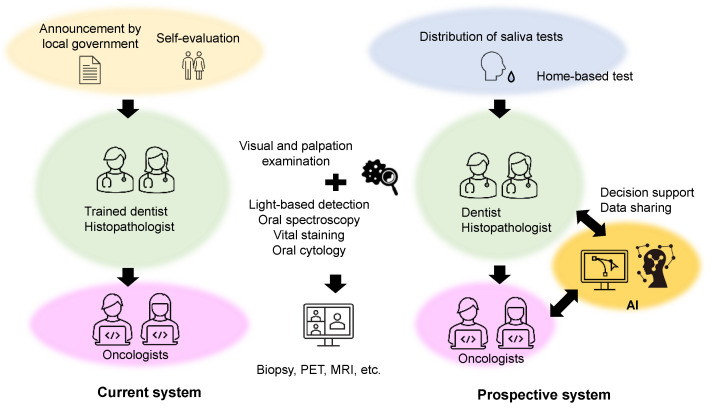
Current and prospective oral cancer screening systems.

**Table 1 metabolites-12-00436-t001:** Summary of salivary metabolomics for OC diagnosis studies.

Analytical Instrument	Discrimination Method or Univariant Statistics	Discrimination Design	Accuracy (AUC or *p*-Value) ^a^	Ref.
RPLC-MS and HILIC-MS	MLR	OSCC from HC	0.997 (Stage I–II)	20
			0.971 (Stage III–IV)	
CE-TOFMS	Wilcoxon rank sum test	OSCC vs. HC	0.00006 ^b^	21
GC-MS	Mann–Whitney test with FDR correction	OSCC vs. HC	3.1755 × 10^−16 c^	22
CPSI-MS	Lasso regression model	OSCC from HC	0.992	23
		PML from HC	0.978	
		OSC from PML	0.917	
UPLC-QTOFMS	MLR	OSCC from HC	0.89	24
		OSFF from OLK	0.97	
CE-TOFMS	MLR	OC from OLP	0.865	25
LC-QTOFMS	ANOVA	OC vs. HC	<0.05 ^d^	26

^a^ The smallest *p*-value at the studies that conducted only univariable statistical analysis. ^b^ Choline, ^c^ Lactose, ^d^ There were 48 significantly different metabolites without the exact *p*-value.

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
