# Peer review of "Salivary Metabolomics for Oral Cancer Detection: A Narrative Review"

_metabolites, 2022, doi:10.3390/metabo12050436_

Round 1

Reviewer 1 Report

The authors submitted a review on salivary metabolomics for oral cancer detection. Most sections are very brief and not always well aligned with the heading such that the manuscripts tends to loose focus (see comments below). However, it is well written otherwise and touches the relevant points of its subject.

Major

Please prepare a table as an overview for all the original manuscripts you have reviewed for this article. Prepare a long version for the SI Material containing information on at least year, authors, DOI, analytical technique (NMR, LC-MS, GC-MS etc), detection mode (non-targeted, targeted), study groups, size of study groups, biomarkers reported, statistical method used (OPLS, PLS-DA, ANOVA, etc.), marker accuracy (or sensitivity/specificity), comments (if any on e.g. clinical relevance, validation, patents, time course assay…). You might consider including a short version of this table (e.g. year, technique, study groups/size) within the manuscript.

L84ff While it is of some value to simply repeat findings (here: biomarkers differentiating OSCC and HC), it should be the aim of a review to also sum up/evaluate the results. Please include statements on the clinical usage of the reported biomarkers as well as on potential overlaps or (if not present) speculate on reasons for the heterogenous results.

L176ff Subheading (“reproducible”) does not match the paragraph content, where only in L186 a hint on the advantage of no additional sample treatment is given. Either adjust subheading or paragraph.

Minor

L44 Isn’t a statement on survival rate similar to one on mortality?

L72 explain color scheme (blue/yellow) of Fig1, check positioning of central indicator lines (blue=bacteria; yellow=molec.comp.?)

L66 Reference #12 is not containing data for the statement (please cite the original source and please ensure that the original source does contain data supporting the statement – at least the abstract of doi: 10.1002/nbm.1369 does not. Otherwise, we prolong citation cascades of unsupported claims.)

L214 I understand Fig.2 more in a way to name influential parameters (relevant for standardization/robust analyses) within the sample processing and would suggest to rephrase the legend accordingly.

Author Response

The authors submitted a review on salivary metabolomics for oral cancer detection. Most sections are very brief and not always well aligned with the heading such that the manuscripts tends to loose focus (see comments below). However, it is well written otherwise and touches the relevant points of its subject.

Major

Please prepare a table as an overview for all the original manuscripts you have reviewed for this article. Prepare a long version for the SI Material containing information on at least year, authors, DOI, analytical technique (NMR, LC-MS, GC-MS etc), detection mode (non-targeted, targeted), study groups, size of study groups, biomarkers reported, statistical method used (OPLS, PLS-DA, ANOVA, etc.), marker accuracy (or sensitivity/specificity), comments (if any on e.g. clinical relevance, validation, patents, time course assay…). You might consider including a short version of this table (e.g. year, technique, study groups/size) within the manuscript.

We are grateful for your suggestion. In accordance with your comments, we have added Table S1, with detailed data. A shorter version of Table 1 was also added (Lines 146-148). Accordingly, the second paragraph of section 2.1 was revised (Lines 96-100).

L84ff While it is of some value to simply repeat findings (here: biomarkers differentiating OSCC and HC), it should be the aim of a review to also sum up/evaluate the results. Please include statements on the clinical usage of the reported biomarkers as well as on potential overlaps or (if not present) speculate on reasons for the heterogenous results.

In accordance with your comments, we have revised the third paragraph of section 2.1 (Lines 121-125).

L176ff Subheading (“reproducible”) does not match the paragraph content, where only in L186 a hint on the advantage of no additional sample treatment is given. Either adjust subheading or paragraph.

In accordance with your comments, we have revised the title of section 3.1 (Line 208).

Minor

L44 Isn’t a statement on survival rate similar to one on mortality?

We are thankful to you for pointing this out. We have revised the sentence (Lines 43-44).

L72 explain color scheme (blue/yellow) of Fig1, check positioning of central indicator lines (blue=bacteria; yellow=molec.comp.?)

To clarify the background color, we edited Figure 1 and revised the figure legend (Lines 74-77).

L66 Reference #12 is not containing data for the statement (please cite the original source and please ensure that the original source does contain data supporting the statement – at least the abstract of doi: 10.1002/nbm.1369 does not. Otherwise, we prolong citation cascades of unsupported claims.)

Thank you for pointing this out. The appropriate references were cited, and we have revised the relevant references in the revised manuscript (Lines 65-67).

L214 I understand Fig.2 more in a way to name influential parameters (relevant for standardization/robust analyses) within the sample processing and would suggest to rephrase the legend accordingly.

In accordance with your comments, we have edited Figure 2 and revised the legend (Lines 302-310).

Reviewer 2 Report

The paper offers insight onto how salivary metabolomics can be used for oral cancer detection.  

The paper discusses briefly diagnostic markers of OC and then describes how there is a need to discriminate between OC and other diseases.  They also discuss biomarkers of other diseases.  Near the end they discuss the importance of standard screening protocols.

I feel the paper lacks organization and thoroughness.  

Thoughts:

  1. They should expand upon the studies where biomarkers have been used to discriminate between OC and HC as well as studies to discriminate OC and other inflammatory diseases
  2. They should not only discuss experimental techniques (LC-MS, NMR, etc) but expand upon the statistical and machine learning techniques used
  3. For the section on establishing standard screening protocols, they need to expand upon the sections

I feel this paper lacks both organization, logical flow, and important detail.  

Author Response

The paper offers insight onto how salivary metabolomics can be used for oral cancer detection.

The paper discusses briefly diagnostic markers of OC and then describes how there is a need to discriminate between OC and other diseases. They also discuss biomarkers of other diseases. Near the end they discuss the importance of standard screening protocols.

I feel the paper lacks organization and thoroughness.

Thoughts:

  1. They should expand upon the studies where biomarkers have been used to discriminate between OC and HC as well as studies to discriminate OC and other inflammatory diseases

Thank you for your suggestion. In accordance with your comments, we have added section 2.5 to describe the comparison of OC biomarkers with those of other inflammatory diseases (Lines 196-206).

  1. They should not only discuss experimental techniques (LC-MS, NMR, etc) but expand upon the statistical and machine learning techniques used

Based on your comments, we have added section 3.2 for describing the results of data analyses (Lines 245-268).

  1. For the section on establishing standard screening protocols, they need to expand upon the sections

Based on your comments, we have revised section 3.3 to describe the protocol in detail (Lines 275-298) and Figure 2 (Lines 301-310).

I feel this paper lacks both organization, logical flow, and important detail.

We are thankful for your insight. In accordance with your comments, we have revised the titles of some sections, such as sections 2.5 (Line 196) and 3.1 (Line 208), and added details, e.g., on analytical instruments (Lines 226-244).

Reviewer 3 Report

This review provides a summary of recent metabolomics applications that have been frequently used to identify and quantify hundreds of metabolites in saliva samples with the objective to identify novel biomarkers associated with various conditions, in oral cancers. The differences in prevalence, epidemiologic characteristics, risk factors, and the current screening programs in India and Japan are discussed. Although the objectives of the review article are of general interest, there are several disadvantages of the current submission. Thus, there are only three figures with examples of rather encyclopedic interest without any figures of NMR and MS metabolomics not even tables of metabolites of interest. Therefore, the review article should be rejected and should be resubmitted after major revision.

Page 6, lines 245-250

Several problems need to be addressed before clinical usage. First, a large-scale valiation should be conducted to evaluate the accuracy of the biomarkers. All published studies on salivary metabolomics included only case-control studies. The prevalence rate of cancer subjects is low in the actual cohort. The positive and negative predictive values, along with sensitivity and specificity should be assessed to prove the screening merit of using salivary metabolomics.”.

The above questions should be addressed in the above review article in the form of tables and figures with the appropriate comments.

Author Response

This review provides a summary of recent metabolomics applications that have been frequently used to identify and quantify hundreds of metabolites in saliva samples with the objective to identify novel biomarkers associated with various conditions, in oral cancers. The differences in prevalence, epidemiologic characteristics, risk factors, and the current screening programs in India and Japan are discussed. Although the objectives of the review article are of general interest, there are several disadvantages of the current submission. Thus, there are only three figures with examples of rather encyclopedic interest without any figures of NMR and MS metabolomics not even tables of metabolites of interest. Therefore, the review article should be rejected and should be resubmitted after major revision.

We are thankful for your insights. We have added a figure in accordance with your comments. A table and a Supplementary Table were added to summarize the findings on salivary biomarkers related to OC diagnosis (Lines 96-100 and 146-149).

Page 6, lines 245-250

Several problems need to be addressed before clinical usage. First, a large-scale valiation should be conducted to evaluate the accuracy of the biomarkers. All published studies on salivary metabolomics included only case-control studies. The prevalence rate of cancer subjects is low in the actual cohort. The positive and negative predictive values, along with sensitivity and specificity should be assessed to prove the screening merit of using salivary metabolomics.”.

The above questions should be addressed in the above review article in the form of tables and figures with the appropriate comments.

In accordance with your comments, we have added Figure 3. In addition, the above sentences were revised by adding two references (Lines 355-363).

Reviewer 4 Report

The paper deals with a literature review of saliva metabolomics used as low- or non-invasive screening test for cancer early detection. Although relatively less used with respect to other biofluids saliva is an ideal substrate containing informative components for disease markers assessment. Therefore, the paper could be interesting for the researchers in the field. Nevertheless, in my opinion, the manuscript organization requires some revision:

1) First of all, the role of saliva with respect to other biofluids metabolomics should be discussed in the introduction with possible references to already published reviews (see for example Angew. Chem. Int. Ed. 2019, 58, 968 – 994 DOI: 10.1002/anie.201804736)

2) Besides the present paragraph organization, the literature description should be arranged and discussed also according to used analytical (Chromatography-MS, NMR,…) and chemometric used methodologies

3) Possibly, a table organising the covered references according to the above methodologies, the specific investigated pathologies and the observed discriminating metabolites should be added to the manuscript for improved readability and use as future reference.

Author Response

The paper deals with a literature review of saliva metabolomics used as low- or non-invasive screening test for cancer early detection. Although relatively less used with respect to other biofluids saliva is an ideal substrate containing informative components for disease markers assessment. Therefore, the paper could be interesting for the researchers in the field. Nevertheless, in my opinion, the manuscript organization requires some revision:

1) First of all, the role of saliva with respect to other biofluids metabolomics should be discussed in the introduction with possible references to already published reviews (see for example Angew. Chem. Int. Ed. 2019, 58, 968 – 994 DOI: 10.1002/anie.201804736)

We thank you for the insightful comments and suggestion for a reference. We have revised the fifth paragraph to discuss the role of saliva, citing an additional study (Lines 65-67 and 71-72). We have also revised the second paragraph of section 3.1 to describe the technical aspects of working with saliva samples by citing the suggested reference (Lines 216-219).

2) Besides the present paragraph organization, the literature description should be arranged and discussed also according to used analytical (Chromatography-MS, NMR,…) and chemometric used methodologies

In accordance with your comments, we have revised the third paragraph of section 3.1 to describe the analytical methods used (Lines 226-244).

3) Possibly, a table organising the covered references according to the above methodologies, the specific investigated pathologies and the observed discriminating metabolites should be added to the manuscript for improved readability and use as future reference.

In accordance with your comments, we have added Table 1 and Supplementary Table 1 to describe the methodology and analyzed metabolites (Lines 96-100 and 146-148).

Round 2

Reviewer 3 Report

The revised version of the review article has been significantly improved, however, in the following sub-section:

Nuclear magnetic resonance (NMR) is the most frequently used method [52]. Compared to MS, NMR has decisive advantages, including higher reproducibility and minimal preparation for any sample type [53]. The pretreatment of the saliva, a viscous liquid, is also simple [54]. This feature is a definitive advantage as it minimizes the chances of causing unexpected errors. This technique has enabled the capturing of a pattern change in salivary metabolomic profiles, i.e., metabolic signature, to distinguish patients with cancer from HCs. Some applications of salivary metabolomics explored using NMR include the detection of OSCC [24,55], salivary gland cancers [56], head and neck squamous cell carcinoma [57,58], and glioblastoma [59]. In addition to cancer, hepatitis B infection 224[56], Parkinson's disease [60], and Alzheimer's disease [61] have been analyzed.”,

it is necessary to include the appropriate Table with the analytes of importance, resonance assignments which are used for identification, detection limits etc. In conclusion, further revision is requested.

Author Response

The revised version of the review article has been significantly improved, however, in the following sub-section:

Nuclear magnetic resonance (NMR) is the most frequently used method [52]. Compared to MS, NMR has decisive advantages, including higher reproducibility and minimal preparation for any sample type [53]. The pretreatment of the saliva, a viscous liquid, is also simple [54]. This feature is a definitive advantage as it minimizes the chances of causing unexpected errors. This technique has enabled the capturing of a pattern change in salivary metabolomic profiles, i.e., metabolic signature, to distinguish patients with cancer from HCs. Some applications of salivary metabolomics explored using NMR include the detection of OSCC [24,55], salivary gland cancers [56], head and neck squamous cell carcinoma [57,58], and glioblastoma [59]. In addition to cancer, hepatitis B infection 224[56], Parkinson's disease [60], and Alzheimer's disease [61] have been analyzed.”,

it is necessary to include the appropriate Table with the analytes of importance, resonance assignments which are used for identification, detection limits etc. In conclusion, further revision is requested.

Response: We thank you for this thoughtful suggestion. In the revised manuscript, we have accordingly added a supplementary table to summarize the biomarkers, saliva collection methods, and NMR methods. We have also revised the second paragraph of 3.1 (Lines 223-227 and 244-246) to cite the relevant data.

(x)English language and style are fine/minor spell check required

Response: We have had the manuscript checked by a native English speaker at a professional editing company to ensure that there are no remaining grammatical or syntax errors. We hope that the manuscript is now suitable for publication in Metabolites

Reviewer 4 Report

The Authors have essentially dealt with the reviewer comments/suggestions

Author Response

(x)English language and style are fine/minor spell check required

Response: We have had the manuscript checked by a native English speaker at a professional editing company to ensure that there are no remaining grammatical or syntax errors. We hope that the manuscript is now suitable for publication in Metabolites